# Novel p-Hydroxybenzoic Acid Derivative Isolated from *Bacopa procumbens* and Its Antibacterial Activity

**DOI:** 10.3390/antibiotics14060591

**Published:** 2025-06-07

**Authors:** Elizabeth Vargas-Anaya, Alejandro Zamilpa, Manasés González-Cortazar, Blanca Eda Domínguez-Mendoza, Ma. Dolores Pérez-García, Minerva Rosas Morales, Ada María Ríos Cortés, Valentin López Gayou

**Affiliations:** 1Departamento de Nanobiotecnología y Biosensores, Centro de Investigación en Biotecnología Aplicada, Instituto Politécnico Nacional (IPN-CIBA), Santa Inés Tecuexcomac 90700, Tlaxcala, Mexico; evargasa1800@alumno.ipn.mx (E.V.-A.); mrosasmo@ipn.mx (M.R.M.); arios@ipn.mx (A.M.R.C.); 2Centro de Investigación Biomédica del Sur, Instituto Mexicano del Seguro Social, Argentina No. 1, Col. Centro, Xochitepec 62790, Morelos, Mexico; gmanases@hotmail.com (M.G.-C.); lola_as@yahoo.com.mx (M.D.P.-G.); 3Laboratorio de RMN, Centro de Investigaciones Químicas, Universidad Autónoma del Estado de Morelos, Av. Universidad 1001, Colonia Chamilpa, Cuernavaca 62209, Morelos, Mexico; bed@uaem.mx

**Keywords:** phenolic acids, antimicrobial resistance, *Bacopa procumbens*, p-hydroxybenzoic acid derivatives, shikimic acid derivatives, plant natural products

## Abstract

**Background/Objectives:** Antimicrobial resistance represents a critical global health challenge that has been exacerbated by the significant decline in antibiotic development. Natural product-based drugs, particularly plant-derived phenolic compounds, offer promising alternatives to conventional antibiotics. This study aimed to isolate and characterize a novel phenolic compound from *Bacopa procumbens*, a Mexican perennial repent plant that is widespread in the Mexican valley and produces a variety of saponins, gastrodin derivatives, and phenolic acids, and to evaluate its antibacterial potential against clinically relevant pathogens. **Methods:** The hydroalcoholic extraction of *B. procumbens* was followed by liquid–liquid partitioning with ethyl acetate. The resulting fraction underwent chromatographic separation and purification. The structural elucidation of the isolated compound was performed using thin-layer chromatography (TLC), high-performance liquid chromatography (HPLC), mass spectrometry (MS-EI), and nuclear magnetic resonance (NMR) techniques. Antimicrobial activity was assessed via a microdilution assay against five bacterial strains, including drug-resistant *Staphylococcus* species and Gram-negative pathogens. **Results:** A novel phenolic compound, 5-(p-hydroxybenzoyl) shikimic acid (5pHSA), was isolated and characterized. The compound demonstrated moderate antibacterial activity against methicillin-resistant *Staphylococcus haemolyticus* and *Escherichia coli* (minimum inhibitory concentration (MIC) = 100 μg/mL) but showed limited efficacy against *Staphylococcus aureus*, MRSA, and *Klebsiella pneumoniae* (MIC > 100 μg/mL). Comparative analysis with the previously isolated compound ProcumGastrodin A revealed structure–activity relationships where the higher lipophilicity of PG-A was correlated with enhanced antimicrobial activity. **Conclusions:** This study establishes 5pHSA as a novel phenolic compound with moderate antibacterial properties. The findings highlight the importance of molecular polarity and structural complexity in determining antimicrobial efficacy, offering valuable insights into the development of phenolic, acid-based antimicrobial agents to address the growing challenge of antimicrobial resistance.

## 1. Introduction

Antimicrobial resistance (AMR) represents a critical global health challenge where pathogenic microorganisms evolve to resist previously effective therapeutic agents [1,2]. This evolutionary process, accelerated by widespread antimicrobial misuse, enables microbes to acquire and transmit resistance genes, causing conventional treatments to become ineffective [3,4,5]. As healthcare institutions experience an increasing challenge from resistant infections [6], the global burden of AMR has reached alarming proportions, directly claiming 1.27 million lives and being associated with 4.95 million deaths worldwide. Six bacteria are responsible for 73% of AMR-attributable deaths: *Escherichia coli*, *Staphylococcus aureus*, *Klebsiella pneumoniae*, *Streptococcus pneumoniae*, *Acinetobacter baumannii*, and *Pseudomonas aeruginosa* [7,8]. Among these, Gram-negative bacteria maintain a critical status, with three specifically classified in this category: carbapenem-resistant Enterobacterales (CRE), third-generation cephalosporin-resistant Enterobacterales (3GCRE), and carbapenem-resistant *Acinetobacter baumannii* (CRAB) [7]. In the Americas, which accounts for 11.1% of AMR-attributable deaths despite representing 13% of the global population, there is a notable prevalence of *Staphylococcus aureus* compared to the global predominance of *Escherichia coli* [9]. Additionally, the region demonstrates particular challenges: higher-income countries primarily struggle with antibiotic overuse, requiring stewardship interventions, while developing nations face fundamental issues of infection control, antibiotic access, and higher mortality rates due to a limited healthcare infrastructure, suggesting that current estimates of AMR’s impact may underestimate its true burden on public health [9,10,11].

Despite the current clinical pipeline containing 97 agents and 16 new approvals since 2017, it remains insufficient to tackle the critical antimicrobial resistance threats [12]. The development landscape is characterized by significant gaps in both existing and emerging treatments: older drugs like colistin and tigecycline present serious toxicity concerns, including nephrotoxicity and neurotoxicity, while newer options such as ceftazidime/avibactam and meropenem/vaborbactam face accessibility barriers due to high costs as well as limited evidence and supporting data [13,14,15]. This therapeutic scenario is intensified by a dramatic decline in antibiotic research investment, evidenced by no major new classes for Gram-negative infections approved between 1962 and 2000, as well as 78% of pharmaceutical companies abandoning antibiotic development since 1990 [1]. The resulting shortage of treatment options complicates the management of increasingly resistant infections and compromises essential medical procedures including organ transplantation, chemotherapy, and invasive surgeries [3].

Natural product-based drugs continue to play a critical role in modern medicine, with approximately half of all small-molecule drugs structurally based on natural products, with higher impacts in areas like bacterial infections (71%) and cancer (65%) [16]. While conventional antibiotic development has declined significantly, plant-derived secondary metabolites have emerged as promising alternatives in addressing this crisis. Plants synthesize a diverse array of bioactive compounds, particularly in response to environmental stressors, which have been used for medicinal purposes throughout human history, with notable examples including morphine from *Papaver somniferum*, paclitaxel from *Taxus brevifolia*, vincristine from *Catharanthus roseus*, and artemisinin from *Artemisia annua* [17,18]. Plant natural products possess unique structural features that distinguish them from synthetic compounds: a larger molecular size, greater structural complexity, higher stereochemical content, richer three-dimensional structure, and lower hydrophobicity, properties that correlate with enhanced binding specificity, reduced preclinical toxicity, and improved clinical trial progression [16,19]. Despite this and the extensive traditional use of medicinal plants worldwide, especially in regions with limited healthcare access, only approximately 15% of global plant species have undergone comprehensive pharmacological, chemical, and toxicological evaluation [18]. The integration of ethnopharmacological knowledge with modern research methodologies presents a promising avenue for discovering novel antimicrobial compounds [20,21]. Recent technological advancements have enhanced the identification and characterization of antibacterial agents from plant sources, offering potential solutions to combat increasing antimicrobial resistance while building upon centuries of traditional medical knowledge [22].

Among these potential compounds, phenolic compounds have demonstrated significant antimicrobial activity as they represent up to 65% of the identified antimicrobial compounds in plant natural products [23]. They are a diverse class of secondary metabolites that are characterized by the presence of at least one aromatic ring attached to a hydroxyl group (a phenolic subunit), and they are categorized according to the number of phenolic subunits linked into the structure, ranging from simple structures to complex polymers [24]. Various phenolic classes show impressive antimicrobial properties like the xanthones from mangosteen pericarps that demonstrate potent activity against Gram-positive bacteria with minimum inhibitory concentration (MIC) values as low as 0.25–1.56 μg/mL; flavonoids like crisoerol and 3′,4′,7-trihidroxiflavona, which exhibit high activity against both Gram-positive and Gram-negative pathogens; and isoflavones, such as lupalbigenina and warangalona, which show strong efficacy against *S. aureus* and methicillin-resistant *Staphylococcus aureus* (MRSA) [23]. Phenolic acids are some of the simplest phenolic compounds. Containing a carboxylic acid group attached to the aromatic ring [24], they exhibit potent antibacterial activity against a wide range of pathogens, including multidrug-resistant bacteria like MRSA and beta-lactamase (BL)-producing *Staphylococcus aureus*, with several compounds showing impressive MIC values (≤10 μg/mL) [25,26]. Their mechanisms of action are multifaceted: they disrupt bacterial cell membranes through hyperacidification and membrane permeabilization, causing leakage of intracellular contents; they inhibit essential enzymes like DNA gyrase, topoisomerase, and d-alanine ligase; and they interfere with bacterial metabolism, block efflux pumps, and alter gene expression [27,28]. Specific phenolic acids like gallic acid and ferulic acid demonstrate significant activity against *Escherichia coli*, *Staphylococcus aureus*, *Listeria monocytogenes*, and *Pseudomonas aeruginosa* by disrupting cell membranes through changes in membrane potential, hyperpolarization, or loss of membrane integrity [25]. Furthermore, their efficacy can be enhanced through structural modifications such as esterification or hydroxylation, making phenolic acids promising candidates for addressing antimicrobial resistance challenges [26].

*Bacopa* is a Plantaginaceae family genus commonly known as water hyssop and it consists of herbaceous plants that grow in aquatic environments and wetlands [29]. The most widespread species of the genus are *B. floribunda* and *B. monnieri*, which are well-known producers of saponins and phenolic compounds with potential pharmacological activity [30,31,32,33]. *Bacopa procumbens*, a Mexican perennial repent plant that is widespread in the Mexican valley and produces a variety of saponins, phenolic acid derivatives, and flavonoids [34,35,36], which have not been completely isolated and characterized. In this work, we described a novel p-hydroxybenzoic acid derivative that was isolated from a hydroalcoholic extract, with the final aim of determining its antimicrobial potential and comparing it with previously reported phenolic acids from the same species. The comparison of the molecules demonstrated that their structural differences influence their lipophilicity and molecular complexity, which, in turn, determines their antimicrobial activity. This opens up the possibility for further structure–activity relationship studies as well as potential structural modifications.

## 2. Results

### 2.1. Ethyl Acetate Fraction Analysis Through HPLC

The chromatogram obtained from an HPLC analysis of the ethyl acetate (EtOAc) fraction is shown in Figure 1. Most of the compounds were grouped in the zone of medium polarity with retention times between 8.2 and 12.6 min, which corresponded to phenolic acid derivatives. Two peaks were particularly prominent: The compound with a t_R_ of 9.25 min corresponded to ProcumGastrodin A (PG-A), which was previously reported by our research group [34], was characterized by two maximum absorption peaks at 215 and 258 nm. Additionally, the compound with a t_R_ of 8.7 min exhibited the same UV absorption peaks as PG-A; however, it demonstrated greater polarity, indicating that they share a similar structure with different spatial arrangements or functional group substituents. This similarity to PG-A led us to proceed with its separation, purification, and structural elucidation, as it was expected to have comparable antimicrobial activity.

### 2.2. Structural Elucidation

After a series of chromatographic runs, a yellow solid was obtained. Analysis using thin-layer chromatography (TLC) identified a compound with a retention factor (R_f_) of 0.52 that absorbed at 254 nm but did not fluoresce at 365 nm (Figure 2c). Furthermore, the derivatization with a Komarowsky reagent revealed the compound as a light pink spot. As previously mentioned, the compound had an HPLC retention time of 8.7 min under the given conditions and its UV-Vis spectrum had two absorption peaks at 198 and 258 nm (Figure 2a,b).

In the mass spectrometry (MS-EI) analysis, the compound of interest showed a negative molecular ion at *m*/*z* 293.11 [M-H]^−^, corresponding to the molecular formula C_14_H_14_O_7_ with an *m*/*z* of 294.07 g/mol. The ^1^H-NMR analysis revealed signals for an aromatic AB system at δ 7.87 ppm (^1^H, d, *J* = 8.6 Hz, H-2′, H-6′) and δ 6.82 ppm (^1^H, d, *J* = 8.9 Hz, H-3′, H-5′), where the carbonyl group was attached on one side (^13^C-NMR at δ 167.5 ppm) and a hydroxyl group (OH) on the other. Additionally, the presence of a double bond was observed in the proton NMR spectrum at δ 6.87 ppm (^1^H, d, *J* = 3.2 Hz, H-2) and in the ^13^C NMR spectrum at δ 138.5 ppm (C-2). In the COSY experiment, H-2 showed coupling with an oxygen-bearing signal at δ 4.44 ppm (^1^H, dd, *J* = 3.5, 3.5 Hz), which was assigned to H-3. The proton of H-3 also showed coupling with another oxygen-bearing signal at δ 3.97 ppm (^1^H, dd, *J* = 4.3, 7.5 Hz), which was assigned to H-4. In turn, H-4 also coupled with an oxygen-bearing signal at δ 5.34 ppm (^1^H, ddd, *J* = 5.3, 7.5, 12.5 Hz), which was assigned to H-5, and this proton (H-5) coupled with two high-field signals at δ 2.89 ppm (^1^H, dd, *J* = 5, 18.2 Hz) and δ 2.38 ppm (^1^H, dd, *J* = 5, 18.2 Hz) that corresponded to a methylene assigned to H-6a and H-6b. In the HMBC experiment, the protons of H-6a and H-6b showed a correlation with three bonds, with a double bond signal at δ 138.5 ppm (C-2), which indicated that this part of the molecule corresponded to a cyclohexene. The connectivity of the carbonyl group containing the aromatic ring showed a correlation with three bonds between C-7′ (δ 167.5 ppm) and the proton of H-5 in the HMBC experiment. All this evidence identified this molecule as 5-(p-hydroxybenzoyl) shikimic acid (5pHSA) (Figure 3). The ^1^H and ^13^C NMR chemical shift data are described in Table 1.

### 2.3. Antimicrobial Activity

Table 2 presents the minimum inhibitory concentration (MIC) values, in μg/mL, of the ethyl acetate (EtOAc) fraction and two isolated compounds (PG-A and 5pHSA) against five bacterial strains: *Staphylococcus aureus* (ATCC 29213), methicillin-resistant *Staphylococcus aureus* (MRSA), methicillin-resistant *Staphylococcus haemolyticus* (MRSH) (derived from ATCC 29970), extended-spectrum beta-lactamase (ESBL)-producing *Klebsiella pneumoniae* (ATCC 700603), and *Escherichia coli* (ATCC 25922). Notably, PG-A and 5pHSA demonstrated distinct antimicrobial profiles. PG-A showed stronger activity against *S. aureus* and MRSH (MIC = 50 μg/mL for both) compared to 5pHSA (MIC > 100 μg/mL and 100 μg/mL, respectively). Interestingly, both compounds demonstrated identical activity against *E. coli* (MIC = 100 μg/mL), while neither showed significant efficacy against ESBL-producing *K. pneumoniae* or MRSA (MIC > 100 μg/mL for both). The EtOAc fraction, from which both compounds were isolated, displayed broader activity against all the tested strains but with higher MIC values (125-500 μg/mL) than the isolated compounds, with the strongest effects against *S. aureus* and *K. pneumoniae* (both 125 μg/mL).

## 3. Discussion

The antimicrobial activity of both compounds (PG-A and 5pHSA) are considered within the moderate activity range (11–110 µg/mL) [26]. Phenolics are the most frequently reported natural products with antimicrobial activity, with flavonoids being particularly prominent due to their ability to disrupt microbial membranes and form complexes with proteins [23,37]. Although phenolic acids typically exhibit lower antimicrobial potency compared to flavonoids like quercetin (MIC = 16–64 µg/mL) or chrysoerol (MIC = 0.06–1 µg/mL) [23], the isolated compounds have a favorable activity range when compared to other phenolic acids such as gallic acid (MIC = 500–2000 µg/mL) and protocatechuic acid (MIC = 200–700 µg/mL) [37,38].

The comparative analysis of their antimicrobial activities revealed interesting structure–activity relationships that could help explain their different efficacies against the tested bacterial strains. The HPLC chromatogram showed that PG-A (t_R_ 9.25 min) is less polar than 5pHSA (t_R_ 8.7 min) despite their structural similarities. 5pHSA presents a simpler structure with a p-hydroxybenzoate group attached to a shikimic acid. In contrast, PG-A contains a more complex structure, sharing the same p-hydroxybenzoate group linked to a shikimic acid moiety skeleton, which suggests that this molecule is a naturally occurring derivative of 5pHSA but has an additional glycosylated cyclohexane core (gastrodin) (Figure 4).

While the number and position of hydroxyl groups influence the antimicrobial activity of phenols, lipophilicity plays a more significant role in determining their effectiveness against bacteria [37,39,40]. Polyphenols with greater hydrophobicity demonstrate enhanced antimicrobial properties because their lipophilicity facilitates stronger interactions between the compound and the phospholipid bilayer of bacterial membranes. Therefore, these compounds can insert themselves between the fatty acid chains in the membrane, destabilizing its structure and compromising its integrity, leading to increased membrane permeability and disruption, altering essential cellular functions, and ultimately triggering bacterial cell death [28,38,41,42]. The higher lipophilicity of PG-A likely facilitates better penetration into bacterial cell membranes, explaining its superior activity against *S. aureus* (MIC 50 μg/mL) and *E. coli* (MIC 100 μg/mL) compared to 5pHSA, which showed limited activity against most strains except MRSH and *E. coli* (100 μg/mL for both).

Additionally, glycosidation can significantly enhance the antimicrobial activity of compounds through multiple complementary mechanisms that improve their therapeutic potential compared to their aglycone counterparts [43]. The addition of sugar residues creates a more amphiphilic structure that enables better recognition and binding to bacterial cell surfaces, enhancing the compound’s ability to disrupt membrane integrity [44,45]. In addition, sugar moieties provide protection against premature degradation, oxidation, and aggregation, thereby extending the compound’s half-life and improving its bioavailability in biological systems [46,47]. The additional sugar moiety in PG-A might improve cellular penetration, combined with enhanced stability and target interaction.

Interestingly, the parent EtOAc fraction demonstrated broader spectrum activity against all the tested strains but with higher MIC values (125–500 μg/mL), suggesting possible synergistic interactions between its constituents. The multiple bioactive compounds in this fraction may target different bacterial cellular components simultaneously, enabling activity against a wider range of pathogens than either isolated compound alone [26,48,49].

These findings highlight how subtle structural modifications significantly influence antimicrobial efficacy and provide valuable insights into the future optimization of these phenolic compounds as potential antimicrobial agents. The knowledge that 5pHSA is a natural precursor to PG-A and is found within the same plant matrix opens the possibility of performing structural modifications on the simpler 5pHSA scaffold to systematically study the role that glycoside incorporation plays in antimicrobial activity, facilitating the establishment of comprehensive structure–activity relationship studies, ultimately allowing researchers to understand the specific contribution of each functional group to the overall antimicrobial potency and mechanism of action.

## 4. Materials and Methods

All the water used in this research, unless otherwise specified, was distilled using a Water Pro/RO system (90750-00, Labconco, Kansas City, MO, USA).

### 4.1. Plant Material

Whole *Bacopa procumbens* plants were collected from a greenhouse at the Centro de Investigación en Biotecnología Aplicada del Instituto Politécnico Nacional at Tepetitla de Lardizábal, Tlaxcala (Mexico), which were previously established using mother plants obtained from San Miguel Regla, Hidalgo (Mexico). The plantlets were identified by Biol. María Edith López Villafranco, and the voucher specimen (1792) was deposited at the Herbarium of Iztacala, Flora útil de México of Universidad Nacional Autónoma de México. The plant material was dried in the dark and ground using an electric mill.

### 4.2. Extraction and Isolation of Ethyl Acetate (EtOAc) Fraction

The ground plant material was extracted with a 50% aqueous ethanolic solution 1:10 ratio *w*/*v* (ethanol 5405, Meyer, Mexico City, Mexico) under reflux conditions for 4 h. The mixture was then centrifuged (Hermle Z323K, Gosheim, Germany) at 3500 rpm for 5 min. The liquid was decanted and concentrated by low pressure distillation using a rotary evaporator (RV8 rotary evporator, HB digital heating bath and RC2 lite recirculating chiller, IKA, Staunfer Germany; V-100 vacuum pump, Buchi, Flawil, Switzerland), and the semi-solid extract was freeze-dried (Labconco 7740021, Kansas City, MO, USA). To isolate an organic fraction, liquid–liquid extraction was performed using an aqueous solution of the extract (1:10 ratio *w*/*v*) and ethyl acetate (Meyer, 1305, Mexico City, Mexico) as the immiscible solvents. The aqueous fraction was concentrated and freeze-dried, while the ethyl acetate (EtOAc) fraction was concentrated until dryness and stored at −4 °C until further analysis.

### 4.3. Chromatographic Separation and Purification of the Metabolite

Approximately 10 g of the ethyl acetate fraction was adsorbed in a mixture of 5 g of silica gel 60 (25–40 mesh, 1.09390.1000, Merck Boston, MA, USA) and 5 g of silica gel 60 RP-18 (40–63 mesh, 1.13900.1000, Merck, Boston, MA, USA), and it was fractionated in a chromatographic gravity column packed with 60 g of silica gel 60 using a solvent gradient of hexane (1480, Meyer, Mexico City, Mexico) and ethyl acetate (1305, Meyer, Mexico City, Mexico) to elute it, which was increased by 5% and volumes of 10 mL were collected. After the analysis of each fraction using thin-layer chromatography, they were recombined based on similarities in chemical composition. The phenolic fraction was sub-fractionated in another normal-phase chromatographic column using a solvent gradient of hexane and ethyl acetate at a ratio of 50:50 (*v*/*v*), with the concentration of methanol increasing by 5% and volumes of 10 mL were collected. Finally, the compound of interest was purified using reverse-phase chromatography and silica gel 60 RP-18 as the stationary phase and a water–acetonitrile gradient that increased by 5%, and 10 mL samples were collected.

### 4.4. Thin-Layer Chromatography Analysis

The less polar fractions obtained from the chromatographic separation were characterized using normal-phase, thin-layer chromatography (TLC): 10 µL of each sample was adsorbed on aluminum sheets of silica gel 60 F254 (1.05735.0001, Merck, Boston, MA, USA), with 0.5 cm spaces between application dots; the chromatographic sheets were eluted using an 85:15 dichloromethane/methanol solution as the mobile phase; once dried, the chromatographic sheets were analyzed under UV light (UV 100Tanon, Shanghai, China) at wavelengths of 254 and 365 nm and derivatized using a Komarowsky (4-hydroxybenzaldehyde (144088, Sigma-Aldrich, Saint Louis, MO, USA) reagent. The polar fractions were analyzed using reverse-phase TLC using the same directions as above for the aluminum sheets of silica gel 60 RP-18 F254S (1.05559.0001, Merck, Boston, MA, USA). These sheets were eluted with a 6:4 water/acetonitrile solution as the mobile phase before the fractions were analyzed and derivatized as the normal-phase fractions.

### 4.5. High-Performance Liquid Chromatography (HPLC) Analysis

HPLC chromatograms were acquired using an HPLC separation module (2695, Waters Corp., Milford, MA, USA) coupled with a photodiode array detector (PDA) (2996, Waters Corp., Milford, MA, USA) using a Discovery C18 column with dimensions of 4.6 mm × 250 mm (internal diameter) and a 5 µm particle size (504971, Sigma-Aldrich, Saint Louis, MO, USA). A binary mobile-phase system was employed, which consisted of (A) an aqueous (AT0110-07, Tecsiquim, Toluca de Lerdo, State of Mexico, Mexico) solution of 0.5% trifluoroacetic acid (TFA) (302031, Sigma-Aldrich, Saint Louis, MO, USA) and (B) acetonitrile (AT0090-7, Tecsiquim, Toluca de Lerdo, State of Mexico, Mexico). The separation was performed using a multi-step gradient elution program: 0% B for 0–1 min; 5% B for 2–3 min; 30% B for 4–20 min; 50% B for 21–23 min; 80% B for 24–25 min; 100% B for 26–27 min; and returning to 0% B for 28–30 min. Throughout the analysis, the flow rate was maintained at 0.9 mL/min, and the injection volume for the samples was 10 µL. The PDA monitored the absorbance across a wavelength range of 190–600 nm; the EtOAc fraction was analyzed at 280 nm and the isolated compound was analyzed at 260 nm.

### 4.6. Ultra-Performance Liquid Chromatography (UPLC) Analysis

The molecular weights of the isolated compound were determined using an Acquity UPLC system (Acquity, Waters Corp., Mildford, MA, USA). The chromatographic setup consisted of a quaternary pump, a temperature-controlled column furnace, and an autosampler equipped with a photodiode array detector connected to a Xevo triple quadrupole mass spectrometer operating at 150 °C. The solvation temperature was maintained at 500 °C, with nitrogen as the solvation gas flowing at 700 L/h. Argon served as the impingement gas at a flow rate of 0.10 mL/min. The analysis was performed on an Acquity UPLC BEH RP-18 column (Waters, Mildford, MA, USA) with a mobile-phase gradient consisting of (A) water (34877, Sigma-Aldrich, Saint Louis, MO, USA) containing 0.5% TFA (302031, Sigma-Aldrich, Saint Louis, MO, USA) and (B) high-purity acetonitrile (AX0156, Merck, Boston, MA, USA), both delivered at a flow rate of 0.3 mL/min. The complete analysis was conducted over a 20 min period.

### 4.7. Antimicrobial Activity

The microdilution method in polystyrene 96-well micro-plates was employed to determine the minimum inhibitory concentration (MIC), following the directions of the M07 manual of the Clinical and Laboratory Standards Institute (CLSI) with some modifications [50,51].

#### 4.7.1. Sample Preparation

Mother solutions of the EtOAc fraction (5 mg/mL), PG-A (1 mg/mL) and 5pHSA (1 mg/mL) were prepared using a 20% dimethyl sulfoxide (DMSO) (472301, Sigma-Aldrich, Saint Louis, MO, USA) solution in sterilized water as the solvent. As an antibiotic control, a stock solution of clarithromycin (1134379, Sigma-Aldrich, Saint Louis, MO, USA) at a concentration of 400 µg/mL was prepared in sterilized water. For the microdilution assay, aliquots of 20 µL of the EtOAc and isolated compound solutions, as well as 2 µL of the antibiotic solution, were taken from these mother solutions.

#### 4.7.2. Microdilution Assay

Two microplates were used for each strain, giving a total of ten microplates. Mueller–Hinton (MH) broth (100 µL; 70192, Merck, Boston, MA, USA) was added to wells B2–G4, B6–G8, and B10–G11 of each plate. Then, 20 µL of the EtOAc mother solution was dispensed into wells B2–B4 of the first five microplates, while 20 µL of the PG-A mother solution was added to wells B6–B8; 80 µL of MH broth was added to these wells to reach a final concentration of 500 µg/mL for EtOAc and 100 µg/mL for PG-A. In the other five microplates, 20 µL of the 5pHSA mother solution and 80 µL of MH broth were added to wells B6–B8 for a final concentration of 100 µg/µL; 2 µL of the clarithromycin mother solution was dispensed into wells B6–B8 along with 98 µL of MH broth for a final concentration of 4 µg/mL. Afterwards, two-fold serial dilutions were made for a total of 6 concentrations: 500, 250, 125, 62.5, 31.5, and 15.62 µg/µL for the EtOAc fraction; 100, 50, 25, 12.5, 6.25, and 3.12 µg/µL for the isolated compounds; and 4, 2, 1, 0.5, 0.25, and 0.125 µg/µL for clarithromycin. Wells B10–G11 were used as the control section; a 20% DMSO solution was added to wells B10–D10, sterilized water was added to wells B11–D11, a bacterial suspension was inoculated into wells E10–G10 as the positive control, and wells E11–G11 were left empty to serve as blanks (Figure 5).

Five bacteria strains, comprising three Gram-positive strains (*Staphylococcus aureus*, methicillin-resistant *Staphylococcus aureus* (MRSA), and methicillin-resistant *Staphylococcus haemolyticus* (MRSH)) and two Gram-negative strains (extended-spectrum beta-lactamase (ESBL)-producing *Klebsiella pneumoniae* and *Escherichia coli*) were tested (Table 3). An inoculum of each strain was prepared by taking two colonies from a plate culture on soy tryptone agar (TSA) (211670, BD, Franklin Lakes, NJ, USA) to prepare a 0.5 McFarland suspension, which was subsequently diluted 1:20 with sterile water; 2 µL of this dilution was added to the wells within 15 min after preparation. The plates were incubated overnight at 37 °C, which exhibited metabolic reduction of 3-(4,5-dimethylthiazol-2-yl)-2,5-diphenyltetrazole bromide (MTT) (T8877, Sigma-Aldrich, Sigma-Aldrich, Saint Louis, MO, USA). The MIC was recorded as the lowest concentration of the compound that could inhibit 100% of bacterial growth.

## 5. Conclusions

The isolation and characterization of 5-(p-hydroxybenzoyl) shikimic acid (5pHSA) from *Bacopa procumbens* contributes to our understanding of phenolic acid derivatives as potential antimicrobial agents. The structural differences between PG-A and 5pHSA appear to significantly influence their antimicrobial efficacy. This structure–activity relationship suggests that lipophilicity and molecular complexity play crucial roles in determining antibacterial potency. Furthermore, the broader antimicrobial spectrum exhibited by the parent EtOAc fraction indicates potential synergistic effects among its constituent compounds. These findings not only expand our knowledge of *B. procumbens* phytochemistry, but also provide valuable insights into the rational design of phenolic, acid-based antimicrobial agents. Future research on the antimicrobial activity of phenolic acids should utilize three interconnected approaches to maximize therapeutic potential. First, continuing the isolation and structural elucidation of phenolic compounds would expand the library of characterized compounds and potentially identify novel bioactive molecules with enhanced antimicrobial properties. Second, conducting comprehensive structure–activity relationship (SAR) studies would provide valuable insights into the molecular features, such as hydroxylation patterns, lipophilicity, glycosidation, and other structural modifications, that confer optimal antimicrobial efficacy, thereby guiding the rational design of more potent derivatives with improved pharmacokinetic profiles. Finally, investigating potential synergistic interactions by combining these phenolic compounds with each other or with conventional antibiotics could reveal promising therapeutic strategies to overcome bacterial resistance mechanisms since these combinations might simultaneously target multiple cellular pathways, leading to more effective antimicrobial formulations with broader spectrum activity.

## Figures and Tables

**Figure 1 antibiotics-14-00591-f001:**
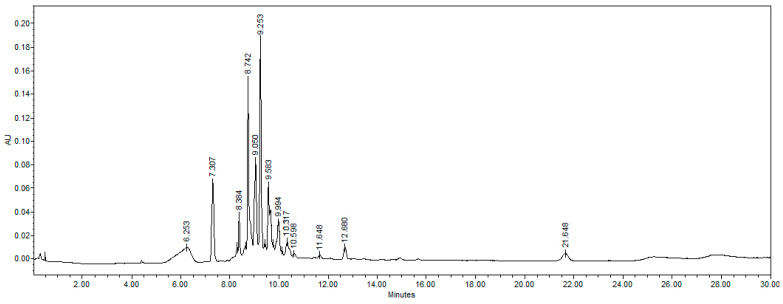
HPLC chromatogram of the ethyl acetate fraction: reverse-phase chromatography, C18 column, gradient elution system consisting of an aqueous solution of 0.5% trifluoroacetic acid and acetonitrile, analyzed at 280 nm.

**Figure 2 antibiotics-14-00591-f002:**
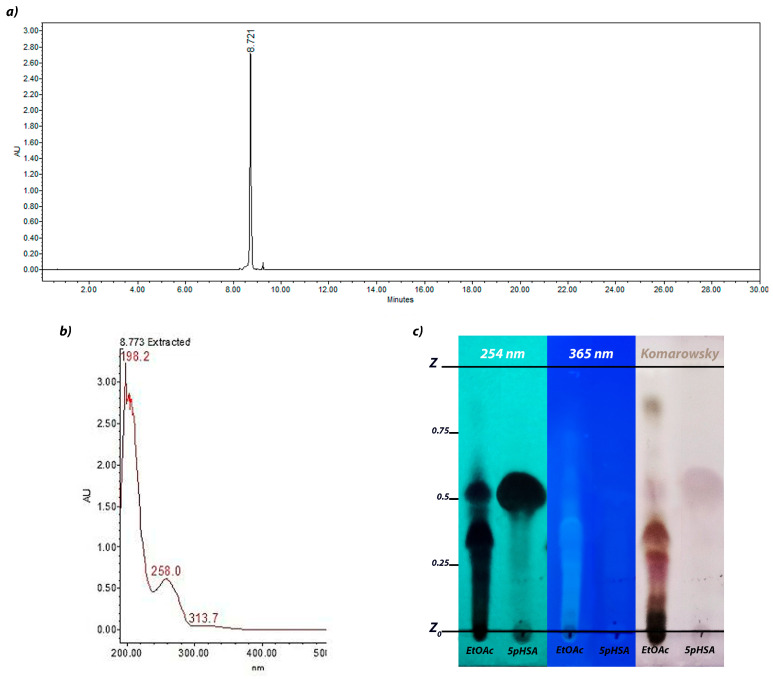
Chromatographic characterization of the isolated compound: (**a**) HPLC chromatogram (reverse-phase chromatography, C18 column, gradient elution system consisting of an aqueous solution of 0.5% trifluoroacetic acid and acetonitrile, analyzed at 280 nm); (**b**) UV-Vis spectra of the isolated compound; (**c**) TLC fingerprint chromatoplate of 5pHSA (silica gel 60 RP-18, water–acetonitrile ratio of 6:4, analyzed at 254 nm and 365 nm, and derivatized with a Komarowsky reagent).

**Figure 3 antibiotics-14-00591-f003:**
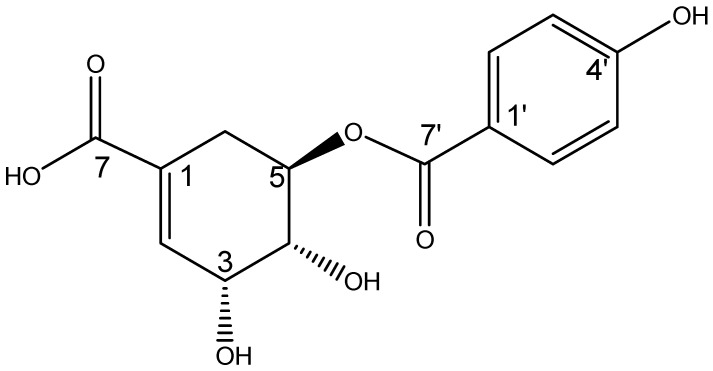
Chemical structure of 5-(p-hydroxybenzoyl) shikimic acid.

**Figure 4 antibiotics-14-00591-f004:**
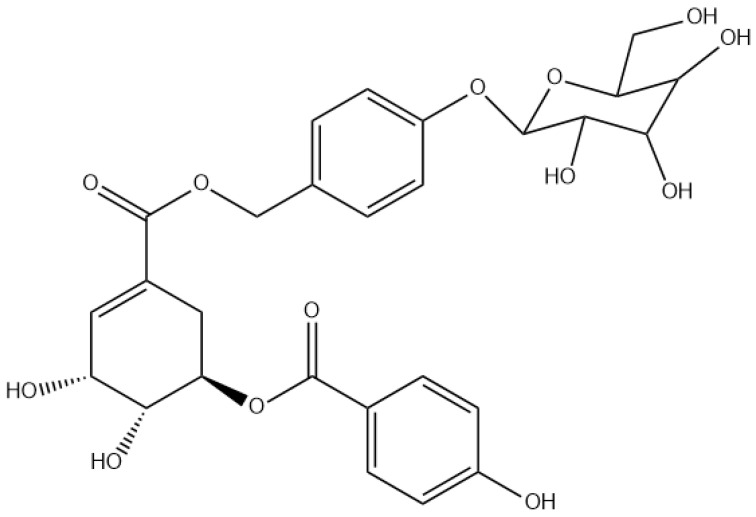
ProcumGastrodin A chemical structure.

**Figure 5 antibiotics-14-00591-f005:**
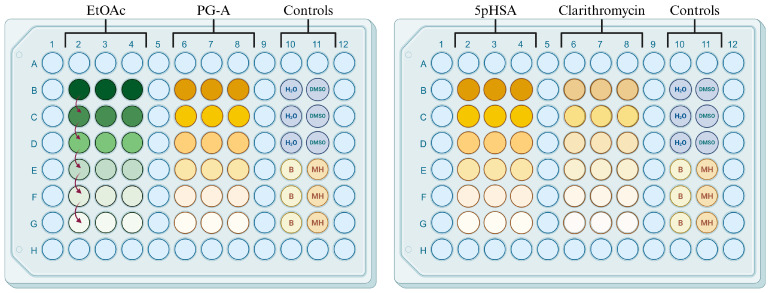
Microdilution assay scheme.

**Table 1 antibiotics-14-00591-t001:** Chemical shift values of 5-(p-hydroxybenzoyl) shikimic acid.

Position	*J* (Hz)	
	δ_H_ (ppm)	δ_C_ (ppm)
1	-	130.7
2	6.87 (1H, d, 3.2)	138.5
3	4.44 (1H, dd, 3.5, 3.5)	67.4
4	3.97 (1H, dd, 4.3, 7.5)	69.9
5	5.34 (1H, ddd, 5.3, 7.5, 12.5)	71.6
6A	2.89 (1H, dd, 5, 18.2)	29.2
6B	2.38 (1H, dd, 5,18.2)	
7	-	170.1
p-hydroxybenzoyl		
1′	-	122.1
2′	7.87 (1H, d, 8.6)	132.8
3′	6.82 (1H, d, 8.9)	116.1
4′	-	163.6
5′	6.82 (1H, d, 8.9)	116.1
6′	7.87 (1H, d, 8.6)	132.8
7′	-	167.5

**Table 2 antibiotics-14-00591-t002:** Antimicrobial activity of the ethyl acetate (EtOAc) fraction and isolated compounds.

Number	Strain	MIC (µg/mL)			
		EtOAc	PG-A	5pHSA	Clarithromycin
1	*Staphylococcus aureus*	125	50	>100	<0.125
2	Methicillin-resistant *Staphylococcus aureus* (MRSA)	250	>100	>100	<0.125
3	Methicillin-resistant *Staphylococcus haemolyticus* (MRSH)	250	50	100	>4
4	ESBL-producing *Klebsiella pneumoniae*	125	>100	>100	<0.125
5	*Escherichia coli*	500	100	100	>4

**Table 3 antibiotics-14-00591-t003:** Bacterial strains tested.

Number	Strain
1	*Staphylococcus aureus* (ATCC 29213)
2	Methicillin-resistant *Staphylococcus aureus* (MRSA) (ATCC 43300)
3	Methicillin-resistant *Staphylococcus haemolyticus* (MRSH) (derived from ATCC 29970)
4	Extended-spectrum beta-lactamase (ESBL)-producing *Klebsiella pneumoniae* (ATCC 700603)
5	*Escherichia coli* (ATCC 25922)

## Data Availability

The original contributions presented in this study are included in the article and Appendix A. Further inquiries can be directed to the corresponding authors.

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
