# Peer review of "Novel p-Hydroxybenzoic Acid Derivative Isolated from Bacopa procumbens and Its Antibacterial Activity"

_antibiotics, 2025, doi:10.3390/antibiotics14060591_

Round 1

Reviewer 1 Report

Comments and Suggestions for Authors

The manuscript titled “Novel p-hydroxybenzoic acid derivatives isolated from Bacopa Procumbens and its antibacterial activity” is an interesting article that emphasizes the need of antibiotic development to overcome antimicrobial resistance. In this work, author identified novel chemical constituent (5pHSA) from Bacopa Procumbens. They also displayed the importance of lipophilicity through performing antimicrobial activity in 5 different strain of bacteria showing compound PG-A, an ester derivative of 5pHSA displayed higher inhibitory activity in few strains. Although antimicrobial activity of the novel isolated compound displayed moderate inhibition, the information obtained from the polarity comparison with the already identified compounds might give some insights in the designing antimicrobial drugs.

The following manuscripts can be accepted for publication in its original form however author have some query and suggestions:

Reviewers Query:

  1. Compound 2 from reference article 34, is the ester derivative of currently identified compound (5pHSA). Reviewers query is that if 5pHSA is the plant derived chemical constituent or the product of ester cleavage of compound 2 from reference article 34 during extraction process.

How author would explain that 5pHSA is not the ester cleaved during extraction under reflux condition.

  1. Reviewer suggests adding compound structure from their prior work (reference article 34) to this article for readers for easier comparison.
  2. Integration in Proton NMR (Fig. S1) would enhance the understanding of compounds protons to the readers

Author Response

Comments and suggestions - Reviewer #1

Authors responses

Compound 2 from reference article 34, is the ester derivative of currently identified compound (5pHSA). Reviewers query is that if 5pHSA is the plant derived chemical constituent or the product of ester cleavage of compound 2 from reference article 34 during extraction process How author would explain that 5pHSA is not the ester cleaved during extraction under reflux condition.

Authors appreciated this observation. To determine if 5pHSA corresponds to a naturally occurring compound, the plant material was extracted by maceration with methanol at room temperature, observing the presence of this compound in the alcoholic extract. Therefore, we conclude that the compound is a natural product.

Reviewer suggests adding compound structure from their prior work (reference article 34) to this article for readers for easier comparison

Chemical structure of ProcumGastrodin A was included in the final version of the manuscript

Integration in Proton NMR (Fig. S1) would enhance the understanding of compounds protons to the readers

New fig. S1 includes the integration of each signal in the 1H NMR experiment.

Reviewer 2 Report

Comments and Suggestions for Authors

This article covers the interesting topics and is well written.

  1. Abstract is very good and comprehensive. The abstract has explained Introduction, Objective, Methods, Results and Conclusions. TLC, HPLC, MS, and NMR, MIC in abstract should be spelled out since these terms appear first.
  2. Introduction is nice, starting from general introduction and ended with specific objective. I suggest authors to highlight the novelty of this study along with gap analysis about this study.
  3. HPLC chromatogram should be accompanied with HPLC condition, otherwise, the authors add the explanation that HPLC condition can be seen in Methods section, since all figures must be self-explaining. Chromatogram in Fig 1 to describe arbutin is well resolved, however, for arbutin peak (tR 6.2) the tailing factor seems severe. 
  4. The results of antibacterial activities did not mention the replication. Please provide the replications along with SD of MIC. In addition, its is recommended to add the statistical test for MIC results
  5. It is very nice if authors compare their results with those reported by others, especially the natural products having the similar chemical composition with Bacopa procumbens.
  6. If applicable, the methods should be accompanied with references

Author Response

Comments and suggestions - Reviewer #2

Authors responses

Abstract is very good and comprehensive. The abstract has explained Introduction, Objective, Methods, Results and Conclusions. TLC, HPLC, MS, and NMR, MIC in abstract should be spelled out since these terms appear first.

This suggestion was applied

Introduction is nice, starting from general introduction and ended with specific objective. I suggest authors to highlight the novelty of this study along with gap analysis about this study.

The contribution of the work and its perspectives were added to the introduction section.

HPLC chromatogram should be accompanied with HPLC condition, otherwise, the authors add the explanation that HPLC condition can be seen in Methods section, since all figures must be self-explaining. Chromatogram in Fig 1 to describe arbutin is well resolved, however, for arbutin peak (tR 6.2) the tailing factor seems severe.

The chromatographic conditions were added

The results of antibacterial activities did not mention the replication. Please provide the replications along with SD of MIC. In addition, its is recommended to add the statistical test for MIC results

The evaluation of the antimicrobial activity was performed in triplicate. The technique is a visual description that only assesses whether or not there is bacterial growth under the given conditions, therefore, a statistical analysis is not applicable

It is very nice if authors compare their results with those reported by others, especially the natural products having the similar chemical composition with Bacopa procumbens.

The observations were considered in the discussion section.

If applicable, the methods should be accompanied with references

All the available references were added to the methods section

Reviewer 3 Report

Comments and Suggestions for Authors

MIC testing results in the manuscript are incomplete and insufficiently reported, which compromises the reproducibility and scientific rigor of the antimicrobial evaluation.

Some concerns:

1. The use of 20% DMSO to dissolve the dry extract from the fraction is very high. Many studies reported that more than 5% of organic solvent used as solvent could  It may exert antimicrobial effects on its own, potentially giving false positive results. Generally, it is recommended to use ≤1–2% DMSO in final assay volume. 

2. While the compound isolated is novel, its antimicrobial activity is weak to moderate (MIC = 100 µg/mL or higher), especially when compared to known phenolic compounds with MICs ≤10 µg/mL. The therapeutic relevance is questionable given the relatively high MIC values

3. Where is the result of MIC of clarithromycin? it is not discussed quantitatively in the results section.

4. MIC end point determination. The manuscript  using  MTT colorant to determine the endpoint of MIC100. I suggest authors to incorporate the microplate images to support MIC interpretation. 

Author Response

Comments and suggestions - Reviewer #3

Authors responses

1. The use of 20% DMSO to dissolve the dry extract from the fraction is very high. Many studies reported that more than 5% of organic solvent used as solvent could It may exert antimicrobial effects on its own, potentially giving false positive results. Generally, it is recommended to use ≤1–2% DMSO in final assay volume

20% DMSO was used to dilute the ethyl acetate fraction and the compounds of interest. The total volume of DMSO in the initial wells for analysis corresponds to 2 microliters, resulting in a starting concentration of 2% DMSO solution, which falls within acceptable limits to avoid false positives. Nevertheless, three control wells containing the DMSO solution were added to each plate to monitor potential interference from the solvent.

2. While the compound isolated is novel, its antimicrobial activity is weak to moderate (MIC = 100 μg/mL or higher), especially when compared to known phenolic compounds with MICs ≤10 μg/mL. The therapeutic relevance is questionable given the relatively high MIC values

Unfortunately, there is no consensus on the MIC range considered to have therapeutic relevance. According to Porras’s classification (reference 26), our compounds are considered to fall within the range of good antimicrobial activity. Although they show moderate activity compared to other phenolic compounds such as flavonoids, they have considerably lower MIC values than other phenolic acids, for which higher MIC values have been reported (200-2000 μg/mL). These data were added to the discussion of results.

3. Where is the result of MIC of clarithromycin? it is not discussed quantitatively in the results section.

The MIC values of clarithromycin were added to the MIC table.

4. MIC end point determination. The manuscript using MTT colorant to determine the endpoint of MIC100. I suggest authors to incorporate the microplate images to support MIC interpretation

The microplate images were added to the supplementary data

Round 2

Reviewer 3 Report

Comments and Suggestions for Authors

1. There is a significant discrepancy in the reported preparation and concentration of test compounds in the MIC assay. For example The EtOAc fraction, it is said to be prepared at 50 mg/mL, and 20 µL is added to 180 µL MH broth. This would yield a final concentration of 5000 µg/mL in the well (2% DMSO). Yet, the reported starting concentration in Table 2 and the method section is 500 µg/200 µL. 

Clarify whether stock solutions were pre-diluted before adding to microplates. clarify also for the PG-A, 5pHSA prepartion, as well as clarithromycin.

2. In the methodology, authors reported the concentration as “µg/200 µL,” which is unconventional and confusing. MIC values should be reported in µg/mL to align with standard clinical and pharmacological guidelines. Indicate this in both text and tables. 

3. The explanation about why PG-A shows stronger activity is clear and makes sense. Still, it would be helpful if the authors could briefly mention whether the glycosylation in PG-A occurs naturally or if it was introduced through modification.

4. It would also be interesting to know whether 5pHSA might be used as a starting structure for future improvements—perhaps by adding similar sugar groups to enhance its activity. More robust discussion on this could help readers understand the potential of 5pHSA as a scaffold for further development.

Minor comment, please check the format again.

  1. please insert space for writing unit (e.g. 280nm --> 280 nm)
  2. consistent punctuation for comma (e..g. 8.7 instead of 8,7 min)
